# Rhodium nanoparticles supported on silanol-rich zeolites beyond the homogeneous Wilkinson's catalyst for hydroformylation of olefins

Yifeng Liu[1,5], Zhiqiang Liu[2,5], Yu Hui[3,5], Liang Wang[1] ✉, Jian Zhang[4], Xianfeng Yi[2], Wei Chen[2], Chengtao Wang[1], Hai Wang[1], Yucai Qin[3], Lijuan Song[3], Anmin Zheng[2] & Feng-Shou Xiao[1,4] ✉

Hydroformylation is one of the largest industrially homogeneous processes that strongly relies on catalysts with phosphine ligands such as the Wilkinson's catalyst (triphenylphosphine coordinated Rh). Heterogeneous catalysts for olefin hydroformylation are highly desired but suffer from poor activity compared with homogeneous catalysts. Herein, we demonstrate that rhodium nanoparticles supported on siliceous MFI zeolite with abundant silanol nests are very active for hydroformylation, giving a turnover frequency as high as ~50,000 h$^{-1}$ that even outperforms the classical Wilkinson's catalyst. Mechanism study reveals that the siliceous zeolite with silanol nests could efficiently enrich olefin molecules to adjacent rhodium nanoparticles, enhancing the hydroformylation reaction.

Hydroformylation is one of the largest industrially homogeneous processes, which convert alkenes into aldehydes and related products using the syngas that can be obtained from coal, biomass, natural gas, and even plastic wastes[1–9]. The current production scale reached more than 24 million metric tons per year, most of which are still strongly dependent on homogeneous Rh catalysts with phosphine ligands[1,2,5–12], which have catalyzed almost half of the industrial hydroformylation processes. The homogeneous nature of these catalysts led to the difficulty in catalyst separation and regeneration as a high-cost process, particularly for that using precious metals. The attempt started with heterogenization of the soluble catalysts[13–16], which improves the catalyst recyclability but requires complex synthesis procedures for grafting the organic ligands on a solid support. The alternative route is

modifying the surface of supported nanoparticle catalysts using excessive phosphine ligands that combined heterogeneous catalyst and organic ligands in the reaction liquor[17–19], but the difficulty in product purification from the solvent containing soluble ligands still limits the wide applications.

In the hydroformylation, the organic ligand-free Rh catalysts have low activity relative to the phosphine-coordinated ones[1]. For example, HRh(CO$_4$) shows a turnover frequency (TOF) at about 9000 h$^{-1}$[1,20], while the TOF of triphenylphosphine coordinated Rh sites [e.g., HRh(CO)(PPh$_3$)$_2$] could reach as high as ~36,000 in the hydroformylation of various olefins[1]. Although the activity was still far to meet the level of triphenylphosphine coordinated catalysts, highly dispersed Rh nanoparticles and even the single-atom Rh on solid

[1]Key Lab of Applied Chemistry of Zhejiang Province and Department of Chemistry & Key Lab of Biomass Chemical Engineering of Ministry of Education and College of Chemical and Biological Engineering, Zhejiang University, Hangzhou 310027, China. [2]National Center for Magnetic Resonance in Wuhan, State Key Laboratory of Magnetic Resonance and Atomic and Molecular Physics and Mathematics, Wuhan Institute of Physics and Mathematics, Innovation Academy for Precision Measurement Science and Technology, Chinese Academy of Sciences, Wuhan 430071, China. [3]Key Laboratory of Petrochemical Catalytic Science and Technology, Liaoning Shihua University, Fushun 113001, China. [4]Beijing Advanced Innovation Center for Soft Matter, Science and Engineering, Beijing University of Chemical Technology, Beijing 100029, China. [5]These authors contributed equally: Yifeng Liu, Zhiqiang Liu, Yu Hui. ✉e-mail: liangwang@zju.edu.cn; fsxiao@zju.edu.cn

carriers provide a new methodology for designing a series of heterogeneous catalysts that can be easily recycled in tests for multiple times. In these cases, the active supports or promoters, such as reducible metal oxide and nitrogen-doped carbon[21–24], are necessary for anchoring and electronically modulating the Rh species. Otherwise, the Rh nanoparticles supported on relatively inert carriers are poorly active. For example, the Rh/SiO2 has TOFs less than 2800 h$^{-1}$ in the hydroformylation of various olefins, such as styrene, 1-hexene, and ethylene[25–28].

Herein, we overturned the general viewpoint by providing a supported Rh nanoparticle catalyst on a siliceous zeolite with superior activity for the hydroformylation, giving a turnover frequency as high as ~50,000 h$^{-1}$, which is significantly higher than that of the general silica supported Rh catalysts under the equivalent test conditions, even outperforms that of triphenylphosphine coordinated Rh catalyst as well-known classical Wilkinson's catalyst. This result demonstrates the great potential for organic ligand-free Rh catalysts in hydroformylation. The key to this success is to employ the siliceous MFI zeolite with abundant silanol nests (S1-OH) within the zeolite crystals, which could enrich the olefins to adjacent Rh nanoparticles to enhance the hydroformylation in toluene solvent. This observation demonstrates the new function of zeolite as a support for metal nanoparticles, which leads to marked gains in the viability of hydroformylation technology using ligand-free and inorganic catalysts.

## Results

### Structure investigation of zeolite with silanols

The proof-of-concept experiment was initially performed using the commercial siliceous MFI zeolite with abundant silanol groups (S1-OH, see SM for details). A series of characterizations including XRD, N2 sorption, TEM, and SEM indicate the typical MFI zeolite structure with uniform crystal sizes, open micropores, and high surface area (Supplementary Figs. 1–4). The silanols on S1-OH zeolite was characterized by the $^1$H MAS NMR experiment (Fig. 1a). Compared to the general silicate-1 (S-1) zeolite, which was synthesized from the classical hydrothermal method using a tetrapropylammonium hydroxide template (Supplementary Fig. 5), S1-OH has a similar silanol concentration according to the $^1$H MAS NMR spectra. However, besides the terminal silanol (-2.0 ppm) mainly located on the external surface of zeolite crystals, S1-OH exhibited additional broad and obvious signal at 3.0–8.0 ppm, which is attributed to the silanols with hydrogen bond interactions. These features are in consistent with the silanol nests that dominantly existed within zeolite crystals. Fig. 1b shows the 2D $^1$H–$^1$H DQ MAS NMR spectrum of S1-OH, showing an obvious correlation signal ranging from 3.0 to 8.0 ppm that was assigned to the silanol nests[29–32]. In contrast, S-1 solely exhibited a relatively weakly correlation signal at -2.0 ppm assigning to the terminal silanols. These results support the different silanols on S1-OH and S-1 zeolite. In one word, both the samples have a similar concentration of silanols but they

dominantly existed as silanol nests on S1-OH and terminal silanol on generally synthesized S-1.

These features were further characterized by $^{29}$Si MAS NMR spectra, where the S1-OH gave signals at −113 ppm and −103 ppm attributing to the typical Si(OSi)4 (Q4) and Si(OSi)3OH (Q3) signals on siliceous zeolite (Supplementary Fig. 6)[31,32]. In the cross-polarization (CP)/MAS NMR spectra, the almost synchronously polarized Q4 and Q3 signals support the presence of abundant silanol nests within zeolite crystals, which have proximity to both Si sites with Q4 and Q3 coordination. In contrast, the general S-1 zeolite showed a similar $^{29}$Si MAS NMR spectrum and silanol concentration (as confirmed by the $^1$H MAS NMR experiments) to that of S1-OH. However, the CP/MAS NMR spectrum exhibited more obviously polarized signals of Q3 than Q4, which is different from the phenomenon on S1-OH. These results suggest that the dominant silanols on the generally synthesized S-1 have proximity to the Si sites with Q3 coordination mostly existing on the zeolite external surface, rather than proximity with the Q4 Si sites within zeolite crystals, in good agreement with the feature of general siliceous zeolite[31,32]. Therefore, it is inferred that the S1-OH and S-1 zeolite have different silanols as nests and isolated silanol, as graphically presented in Supplementary Fig. 7.

The Rh nanoparticles were loaded on the S-1-OH and S-1 zeolites from an impregnation method, giving the Rh/S1-OH and Rh/S-1 samples with Rh loading at ~0.06 wt% (Supplementary Table 1). The zeolite structure with open micropores was well maintained after loading Rh (Supplementary Figs. 8–10). Fig. 1c showed TEM image characterizing the Rh/S1-OH sample, giving uniform distribution of Rh nanoparticles on the zeolite with an average nanoparticle size of 2.0 ± 1.1 nm (Supplementary Fig. 3). To identify the location of Rh nanoparticles on the S1-OH, we cut the zeolite crystals into slices and then performed the TEM characterization, which could minimize the influence of overlapped imaging Rh nanoparticles and zeolite crystals. As shown in the TEM image in Supplementary Fig. 10, the zeolite region was free of Rh nanoparticles, and the observed Rh nanoparticles are on the zeolite external surface. These data confirm that the Rh nanoparticles indeed existed on the external surface of zeolite crystals on the Rh/S1-OH sample. Because the S1-OH support was siliceous without ionic framework as aluminosilicate zeolite, the isolated or binuclear Rh sites would not exist on the Rh/S1-OH sample. The Rh dispersions (fraction of accessible Rh sites to the total amount of Rh atoms) were similar (~25%) for Rh/S1-OH and Rh/S-1, where the Rh species had similar electronic statues as supported by the Rh 3d XPS spectra (Supplementary Figs. 11 and 12).

### Catalytic performance in hydroformylation

The catalytic evaluation of the supported Rh catalysts was performed in the hydroformylation of styrene with syngas (molar ratio of CO/H2/Ar at 45/45/10) in toluene solvent (Fig. 2a). Under the given reaction conditions (3 MPa of syngas, 110 °C, 4 h), the pure zeolite without Rh

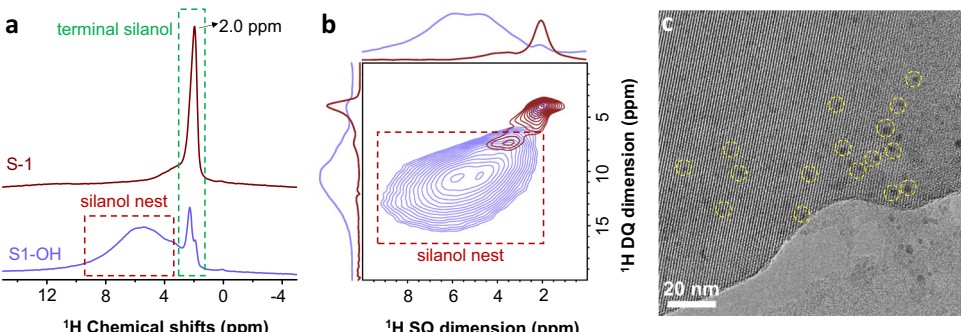

**Fig. 1 | Structure investigation. a** $^1$H MAS NMR spectra of S1-OH and S-1 zeolites. **b** 2D $^1$H-$^1$H DQ MAS NMR spectra of the S1-OH (purple line) and S-1 (red line) samples. **c** TEM image of Rh/S1-OH.

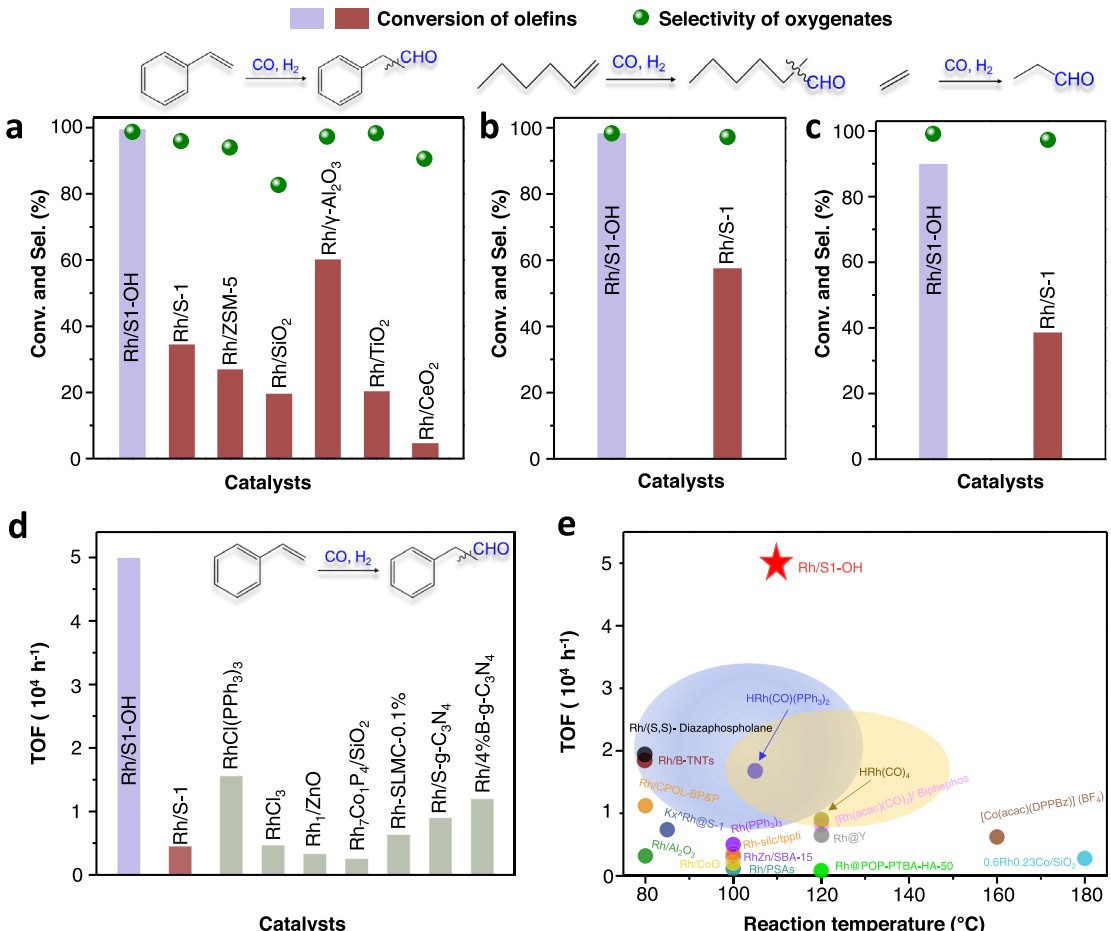

**Fig. 2 | Catalytic performance in hydroformylation. a** Data showing the performances of various catalysts in the hydroformylation of styrene. **b**, **c** Data showing the performances of Rh/S1-OH and Rh/S-1 catalysts in the hydroformylation of hexene and ethylene. Reaction conditions: syngas with a molar ratio of CO to H$_2$ at 1 (molar ratio of CO/H$_2$/Ar at 45/45/10), 30 mg of catalyst, 2.5 mmol of styrene and 1-hexene, 5 mL of toluene as solvent, butanol as internal standard, 110 °C, 4 h. For ethylene hydroformylation, 0.5 MPa of ethylene and 2.4 MPa of syngas were employed. 0.1 MPa of methane was used as an internal standard. The carbon balances were over 99.5% for all the tests. **d** TOF comparison between Rh/S1-OH and different Rh catalysts tested previously in the hydroformylation of styrene. These tests were performed under comparable reaction temperatures of 100–110 °C. The details for the catalysts tested previously are summarized in Supplementary Table 2. **e** TOF comparison between Rh/S1-OH and Rh catalysts tested previously in the hydroformylation of LAOs. The details for the catalysts tested previously are summarized in Supplementary Table 2.

species failed to produce oxygenate products, while the zeolite-supported Rh nanoparticles were very active for the hydroformylation. For example, the Rh/S1-OH showed a styrene conversion at 99.6% with selectivity to phenylpropyl aldehyde at 98.7% (molar ratio of linear to branched products at 1.37), and a slight amount of ethylbenzene was detected with selectivity lower than 1.5% (Supplementary Fig. 13). Under the equivalent reaction conditions, the Rh/S-1 showed styrene conversion at 34.3%, which is much lower than that of Rh/S1-OH. Notably, the Rh/S-1 exhibited relatively higher selectivity to undesired ethylbenzene (4.0%) because of the over hydrogenation. The Rh nanoparticles loaded on the aluminosilicate zeolite of ZSM-5 (Rh/ZSM-5, Fig. 2a and Supplementary Fig. 14) showed styrene conversion at 26.8% with phenylpropyl aldehyde selectivity at 94.0%. Considering these zeolite-supported catalysts have similar loadings and nanoparticle sizes, the significantly different performances persuasively confirm the crucial role of zeolite support for the reactions. The amorphous silica supported Rh nanoparticles (Rh/SiO$_2$, Supplementary Fig. 15) showed a very low conversion at 19.5% with phenylpropyl aldehyde selectivity at 82.7%. The other catalysts of metal oxides supported Rh nanoparticles such as Rh/γ-Al$_2$O$_3$, Rh/TiO$_2$, and Rh/CeO$_2$ catalysts (Supplementary Figs. 16 and 17), exhibited styrene conversions at 60.0%, 20.2%, and 4.5%,

respectively. These data demonstrate the unusual catalytic performances of the Rh/S1-OH for hydroformylation.

We evaluated the TOF of the Rh/S1-OH catalyst in the hydroformylation of styrene, giving an average reaction rate based on Rh sites as high as ~12,500 mol mol$_{Rh}^{-1}$ h$^{-1}$. The styrene conversion was controlled to be lower than 20% for getting a persuasive evaluation of the activity. According to the Rh dispersion, the TOF of Rh sites could reach ~50,000 h$^{-1}$. The RhCl$_3$ without organic ligand and support showed the TOF at 4700 h$^{-1}$. The 0.006%Rh$_1$/ZnO which was reported as a highly efficient catalyst for styrene hydroformylation, gave a TOF at 3333 h$^{-1}$ (according to the given TON in ref. 22) under comparable reaction conditions (Fig. 2d and Supplementary Table 2). The Wilkinson's catalyst [RhCl(PPh$_3$)$_3$, Supplementary Fig. 18], known as a classical homogeneous catalyst for hydroformylation[2], exhibited a TOF at ~15,600 h$^{-1}$ under the equivalent reaction conditions (Fig. 2d). These data indicate the superior catalytic activity of the Rh/S1-OH catalyst, even outperforming that of the homogeneous catalysts[1,5,8–10]. Notably, we understand that the reactions tested previously were performed under varied reaction conditions, it is a challenge to obtain the data on TOFs of various catalysts under the equivalent tests. Therefore, these data for hydroformylation of styrene under comparable reaction temperatures of 100–110 °C were shown here for comparison[22,33–35].

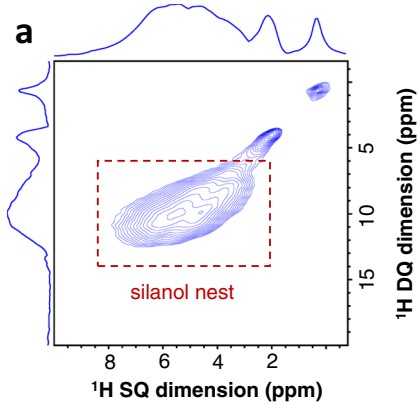

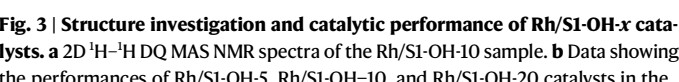

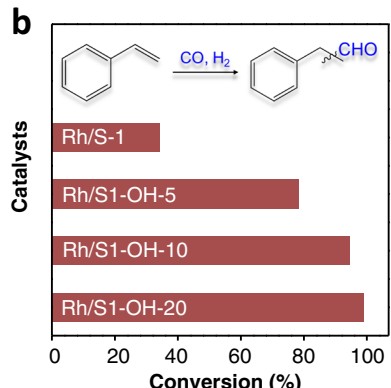

**Fig. 3 | Structure investigation and catalytic performance of Rh/S1-OH-x catalysts. a** 2D $^1$H−$^1$H DQ MAS NMR spectra of the Rh/S1-OH-10 sample. **b** Data showing the performances of Rh/S1-OH-5, Rh/S1-OH−10, and Rh/S1-OH-20 catalysts in the hydroformylation of styrene. The reaction conditions are the same to those in Fig. 2a.

In the hydroformylation of other olefins such as 1-hexene, a similar trend was also observed. The Rh/S1-OH showed 1-hexene conversions at 98.4% with 98.3% selectivity to the corresponding aldehydes (Fig. 2b). In contrast, the 1-hexene conversion was only 57.6% on the Rh/S-1 catalyst. An additional attempt was performed in the hydroformylation of ethylene, an important process for obtaining propanal from the basic petrochemicals. As shown in Fig. 2c, the Rh/S1-OH showed the ethylene conversion at 90.0% with propanal selectivity at 99.1%. In contrast, the Rh/S-1 gave a much lower ethylene conversion at 38.6% with propanal selectivity at 97.3%. Fig. 2e shows the data characterizing the TOFs of different catalysts tested previously in the hydroformylation of linear alpha-olefins (LAOs) under the optimized reaction conditions for each catalyst. Clearly, the Rh/S1-OH still exhibited significant advance compared with other catalysts, even including the homogeneous catalysts.

The substrate scope was extended to different olefins (Supplementary Table 3). The substituted styrene with methyl and chloro groups at *para*-position (*p*-methylstyrene and *p*-chlorostyrene) exhibited excellent activity and selectivity in the hydroformylation reaction over the Rh/S1-OH catalyst, giving conversions of 99.0% and 95.0% with oxygenate selectivities of 98.2% and 99.0%, respectively. The methyl groups at *meta*-position do not obviously influence the reaction, giving *m*-methylstyrene conversion at 97.0%. The *ortho*-methylstyrene has a stronger steric hindrance than *meta*- and *para*-methylstyrene in the hydroformylation reactions, but a high conversion at 92.0% was still obtained under the equivalent test conditions. 2-phenyl-1-propene is a typical molecule with an even larger steric hindrance on the C=C bond, but the Rh/S1-OH still showed 61.7% conversion with 99.0% selectivity to 2-phenylpropionaldehyde in 24 h. Prolonging the reaction time to 36 h could further increase the 2-phenyl-1-propene conversion, reaching 83.4% with 2-phenylpropionaldehyde selectivity at 96.5%. These results confirm the high efficiency of the Rh/S1-OH in the hydroformylation of different olefins.

After each reaction run, the Rh/S1-OH catalyst can be easily filtered, washed with toluene, and reused in the next run. In the recyclability tests, the conversions were controlled to be ~85% by shorting the reaction time compared with the standard test. Because of the physical loss of catalyst during the experimental operation for each recycle, we presented the profiles of styrene conversion as a function of catalyst amount in each run (Supplementary Fig. 19). Clearly, the styrene conversion was linearly consistent with the catalyst amount during the recycle tests, suggesting good recyclability. Supplementary Fig. 20 showed the TEM image characterizing the used Rh/S1-OH catalyst after the recycle tests, exhibiting the uniformly distributed Rh nanoparticles with an average size of 2.2 ± 1.1 nm, which was almost unchanged compared with the fresh catalyst. The used Rh/S1-OH had a

Rh loading amount at ~0.06 wt% that is similar to the as-synthesized catalyst (Supplementary Table 1). To further investigate the heterogeneous features of Rh/S1-OH catalyzed hydroformylation, we performed a hot filtration test by separating the catalyst from the reaction liquor after a primary reaction for 2 h under the standard test conditions (styrene conversion at 65.6%). Then, the resulted liquor was further used in another test by re-feeding the syngas but without adding catalyst for another 2 h, resulting in the styrene conversion at 66.0%, which suggests the switched-off styrene conversion within error bounds after removal of the catalyst (Supplementary Fig. 21). These results confirm the leaching and sintering resistance of the Rh/S1-OH catalyst during hydroformylation.

## Catalysts with varied internal silanol concentration

To further understand the crucial role of zeolite with silanol nests in hydroformylation and exclude the influence of the possible unknown impurities in the commercial zeolite, we reasonably synthesized siliceous MFI zeolites with artificially induced silanol nests (see SM for details, Supplementary Figs. 22–26). The zeolite was synthesized using an organosilane-assisted strategy, and these organic groups could be transferred into silanols via calcination in air[30]. Following this strategy, the Rh/S1-OH-10 sample was obtained with the organosilane concentration to the total amount of silica in the starting gels at 10% and then loading Rh with amount at 0.06 wt% (Supplementary Figs. 23b, 24b, and 25b). $^1$H MAS NMR spectra of these samples also showed broad signals at 3.0–8.0 ppm, Meanwhile, obvious correlation signals ranging from 3.0 to 8.0 ppm were observed in the 2D $^1$H−$^1$H DQ MAS NMR spectrum of Rh/S1-OH-10 (Fig. 3a and Supplementary Fig. 27), confirming the presence of silanol nests within zeolite crystals induced by the organosilane-assisted method. This feature is further confirmed by the $^{29}$Si NMR and FTIR spectra (Supplementary Fig. 28). By adjusting the organosilane amount in the starting gel, the Rh/S1-OH-5 and Rh/S1-OH-20 with dominant silanol nests but different concentrations could be obtained (Supplementary Figs. 22–27).

We evaluated these catalysts in the hydroformylation of styrene under the standard reaction conditions except for shorting the reaction time to get uncompleted conversions for providing a persuasive comparison. As shown in Fig. 3b, the Rh/S1-OH-5, Rh/S1-OH-10, and Rh/S1-OH-20 showed styrene conversions at 78.3%, 94.6%, and 98.0%, respectively. Considering similar rhodium loading, nanoparticle sizes, and electronic state of Rh nanoparticles in these samples, these different catalytic activities strongly indicate the positive relationship between silanol concentrations and catalytic activity of the Rh/S1-OH catalysts. In these cases, the Rh/S1-OH-10 and Rh/S1-OH-20 exhibited comparable catalytic performances to that of the Rh/S1-OH based on the commercial silanol nest-rich S-1 zeolite. The Rh/S1-OH-5, even with

relatively fewer silanol groups, still exhibited an improved performance compared with the general Rh/S-1 catalyst.

To further strengthen the importance of silanol nests to the catalysis, we reasonably prepared the desired siliceous MFI zeolites by degallation of the parent Ga-MFI zeolite with different Si/Ga ratios at 30 and 60, obtaining the zeolite samples with Si/Ga ratio higher than 1000 (Ga content lower than 0.012 wt%). The degallation would lead to the silanol nests with concentration linearly related to the gallium content[36]. By loading Rh on these zeolites, the obtained Rh/MFI-deGa-30 and Rh/MFI-deGa-60 catalysts showed styrene conversions at 95.0% and 80.7% in the hydroformylation, which was obviously higher than 34.3% over Rh/S-1 under the same reaction conditions (Supplementary Figs. 29 and 30). In addition, the Rh/MFI-deGa-30 is more active than Rh/MFI-deGa-60, supporting the crucial role of silanol nests in catalysis.

## Mechanism study

The Rh-catalyzed hydroformylation processes have been extensively investigated (Supplementary Fig. 31)[18,19,37-42], and the olefin-related steps are usually crucial, particularly for the supported Rh nanoparticles. In the kinetic studies for styrene hydroformylation, the Rh/S-1 zeolite catalyst showed apparent reaction orders to styrene, $H_2$, and CO at -1.10, -0.53, and -0.88, respectively (Fig. 4a–c). These results are slightly different from those in the gas-phase hydroformylation, which might be due to the hindered access to the active site in the toluene solvent. For example, the kinetic reaction order for CO in the gas-phase reaction is usually negative because of the strong adsorption of CO to the Rh sites, but the positive reaction orders were obtained in the reaction in solvents[43-45]. In the reaction with a constant syngas pressure, it is reasonable to understand that a higher concentration of olefins close to the active sites would accelerate the reaction. For the

Rh/S1-OH catalyst, the kinetic reaction order to styrene appeared at -0.15 (Fig. 4a), which was quite different from that of the supported Rh nanoparticle catalysts and the homogeneous ligand-containing catalysts tested previously[18]. This close-to-zero reaction order to olefins confirms the insensitivity of olefin concentration to the reaction rate, which might be due to the enriched olefin molecules around the Rh nanoparticles of the Rh/S1-OH catalyst (Supplementary Fig. 32). In this case, the reaction orders to $H_2$ and CO are -0.82 and -1.01, respectively (Fig. 4b, c). Similar results were obtained in the kinetic studies in the hydroformylation of 1-hexene and ethylene, where much lower kinetic reaction orders to olefins were obtained over Rh/S1-OH catalyst (-0.25 and -0.13) than that over Rh/S-1 catalyst (-0.90 and -1.18, Supplementary Fig. 33).

The effect of silanol nests on olefin sorption was explored by multiple techniques. Supplementary Fig. 34 shows the temperature-programmed desorption test of ethylene on the Rh/S1-OH and Rh/S-1 catalysts, where the Rh/S1-OH exhibited a stronger desorption signal at a higher temperature compared with Rh/S-1, suggesting the improved ethylene adsorption capacity and strength on the Rh/S1-OH. Fig. 4d gives the ethylene-adsorption FTIR spectra of the Rh/S1-OH catalyst, showing obvious signals at 3742, 3724, and 3300−3550 $cm^{-1}$ that are assigned to the terminal silanol (3742 $cm^{-1}$) and the silanol nests (3724 and 3300−3550 $cm^{-1}$), respectively[46,47]. Pulsing ethylene to the sample resulted in an obvious red shift of the broad signal 3300−3550 $cm^{-1}$. Simultaneously, the signal at 3724 $cm^{-1}$, which was obviously stronger than the terminal silanol on the fresh sample, was continuously decreased with much weaker intensity than the terminal silanol after inducing ethylene. These data suggest the interaction between the silanol nests with the ethylene molecules, while the terminal silanols that usually existed on the zeolite external surface were almost

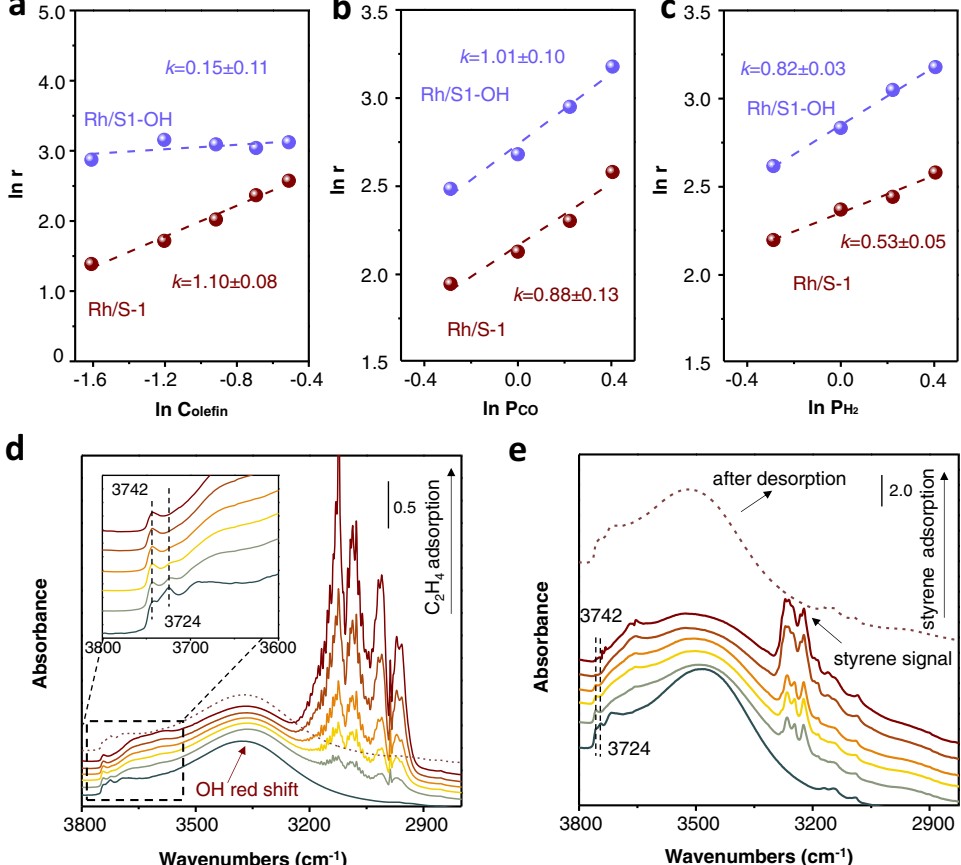

**Fig. 4 | Mechanism investigation.** Kinetic reaction order to **a** styrene, **b** CO, and **c** $H_2$ in the hydroformylation of styrene over Rh/S1-OH and Rh/S−1 catalysts. **d**, **e** In situ ethylene and styrene adsorption FTIR spectra of Rh/S1-OH. Inset D, enlarged view of the signals at 3600–3800 $cm^{-1}$.

unchanged during the ethylene adsorption. Similar results were obtained in the styrene and 1-hexene adsorption FTIR test (Fig. 4e and Supplementary Fig. 35). The olefin molecules in zeolite could be removed under vacuum, regenerating the FTIR spectrum similar to that of the fresh Rh/S1-OH.

These data above confirm the efficient adsorption of olefin molecules by the siliceous MFI zeolite with abundant silanol nests, which reasonably enrich the olefin molecules around the Rh nanoparticles to accelerate the catalysis. This effect was further explored by a molecular dynamics (MD) simulation at 110 °C using ethylene as a model. These molecules were homogeneously distributed in the system without zeolite (Supplementary Fig. 36), but they diffused rapidly and adsorbed in the zeolite micropores with the existence of silanol-rich MFI (S1-OH) zeolite (Fig. 5a, Supplementary Fig. 36, Supplementary Table 4). As a result, a quantitative equilibrium with 89.0% of the ethylene molecules was enriched in the S1-OH zeolite (Fig. 5b and Supplementary Fig. 37). The olefin enrichment effect was also supported by the average distance between the neighboring olefin molecules, where the ethylene molecules have much closer distances in the zeolite relative to that in the zeolite-free phase (Supplementary Figs. 38 and 39).

The function of silanol nests was performed by density functional theory (DFT) calculations, showing the adsorption energy for ethylene molecule at −0.59 eV, which indicates stronger adsorption relative to −0.38 eV in the silanol-free zeolite micropores (Fig. 5c, Supplementary Figs. 40 and 41). A similar trend was also observed in the adsorption of 1-hexene (Supplementary Table 5), showing stronger adsorption at the silanol nest relative to the silanol-free micropores. This feature should be due to the hydrogen-π interaction between the silanol nest with olefin molecule[46]. The diffusion coefficient ($D_s$) of ethylene molecules in the silanol-rich zeolite was of $4.1 \times 10^{-9}$ m$^2$/s at 110 °C (Supplementary Fig. 42), suggesting the movable olefins in the zeolite that benefits their access to the Rh nanoparticles for the catalysis. In the dynamic model, the number of olefin molecules as a function of distance around the Rh nanoparticles was explored (Fig. 5d, e), exhibiting a significant enrichment of olefins around the Rh nanoparticle supported on zeolite relative to that in the homogeneous phase. For example, in a space with a radius of 30 Å around the Rh nanoparticles, there were about 6 and 33 ethylene molecules around Rh in the homogeneous phase and on the S1-OH zeolite surface, respectively (Fig. 5e). By analyzing the distribution of the olefins with high density around the Rh nanoparticle on the zeolite, 91.3% of them were in the zeolite crystals and on the zeolite surface, while only 8.7% in the free system (Fig. 5d). These results showed the effect contributed by zeolite support that could accelerate the interaction between olefin molecules and the Rh sites. Generally, the kinetic reaction order is a function of the transforming rate to the coverage on the catalyst surface. Such olefin enrichment would reasonably reduce the kinetic reaction order from -1.10 on the Rh/S1 catalyst to close-to-zero on the Rh/S1-OH catalyst. Meanwhile, the CO transformation and adsorption might not be influenced by the different zeolite support, thus leading to similar apparent kinetic reaction orders on both catalysts.

Furthermore, we performed the hydroformylation of bulky 2,4,6-trimethylstyrene over the Rh/S1-OH. Because of the impregnation method for the preparation, the Rh nanoparticles were dominantly loaded on the external zeolite surface. Therefore, the 2,4,6-trimethylstyrene should directly access the Rh nanoparticles on the surface of the Rh/S1-OH, but the S1-OH zeolite cannot enrich the bulky 2,4,6-trimethylstyrene because the size of zeolite micropores is less than the diameter of 2,4,6-trimethylstyrene. As a result, the Rh/S1-OH catalyzed the hydroformylation of 2,4,6-trimethylstyrene with conversion at 48.6%, which is much less than that (72.1%) of styrene hydroformylation over the same catalyst. In contrast, the generally supported rhodium catalyst such as Rh/SiO$_2$, showed similar conversion for hydroformylation of styrene and 2,4,6-trimethylstyrene (19.5% vs. 22.0%, Supplementary Fig. 43). These results support the crucial role of

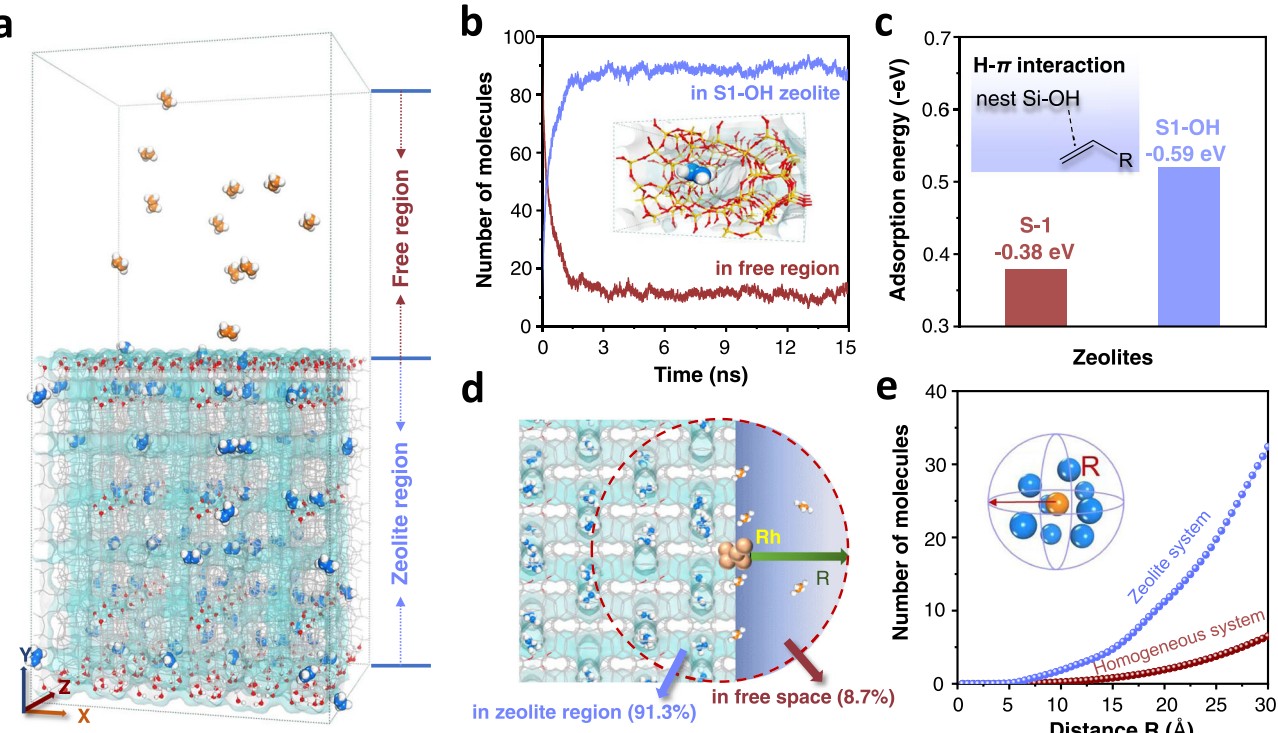

**Fig. 5 | Theoretical calculation. a** Distribution of ethylene molecules (orange-white or blue-white) in the S1-OH zeolite system (gray framework with hydroxyl groups in red-white). **b** Number of ethylene molecules in free space and zeolite during the diffusion process. **c** Adsorption structure and energy of ethylene in S1 and S1-OH zeolites. **d** Scheme showing the olefin enrichment around the supported Rh nanoparticle on zeolite. **e** The number of ethylene molecules as a function of distance to the Rh nanoparticle in homogeneous system and zeolite system simulating the homogeneous Rh catalyst and Rh/S1-OH catalyst, respectively.

zeolite with silanol groups for accelerating the hydroformylation of olefins with appropriate molecular sizes. Another test was performed by loading Rh nanoparticles on the zeolite-containing organic template (as-synthesized zeolite without calcination), where the micropores were blocked to eliminate the enrichment effect. As a result, this catalyst exhibited remarkably decreased activity compared with the Rh nanoparticles on silanol nest-rich zeolite with open micropores (Supplementary Fig. 44). These data, again, confirm the importance of zeolite micropores with silanol nests for hydroformylation.

In sum, we have demonstrated efficient zeolite-supported Rh nanoparticle catalysts with superior activity for hydroformylation. Key to the success is to induce abundant silanol groups within the zeolite crystals, which efficiently enriched and shipped the olefin substrates to boost the hydroformylation, as evidenced by the experimental and theoretical studies. Owing this feature, a recorded catalytic activity of the supported Rh catalyst was achieved to outperform the previous heterogeneous Rh nanoparticle catalysts and even the classical homogeneous Wilkinson's catalyst. Some Rh-containing zeolite catalysts have been previously developed for hydroformylation reactions, they relied on the encapsulation of Rh cations or nanoclusters within the zeolite micropores[48,49]. In contrast, this work represents a large step forward compared with these works and confirms the great efficiency of rationally-designed zeolite in promoting these reactions. Future work should therefore focus on further exploring new functions of zeolite in hydroformylation that would benefit the development of more efficient heterogeneous catalysts.

## Methods
### Materials
All reagents were commercially obtained without purification. Tetraethyl orthosilicate (TEOS), tetrapropylammonium hydroxide (TPAOH, 40 wt%) and ammonia solution (25–28 wt%) were supplied by Shanghai Cairui Chemical Technology Co. Ltd. Rhodium chloride hydrate was obtained from Beijing HWRK Chem Co. Ltd. $SiO_2$, γ-$Al_2O_3$, styrene, 1-hexene, 1-octene, 4-methylphenylene, 4-chlorostyrene, alpha-methylstyrene, $Ga(NO_3)_3 \cdot xH_2O$, and diethoxydimethylsilane (DEMS) were obtained from Aladdin Chemical Reagent Company. Commercial silicate-1 zeolite (denoted as S1-OH in this work) was obtained from Nankai University Catalyst Co. Ltd.

### Catalysts preparation
Synthesis of S-1 zeolite. 3 mL of TPAOH (40 wt%) and 3.5 g of TEOS were mixed in 11.28 g of water, stirring at room temperature for 6 h. Then, the mixture was transferred into an autoclave and hydrothermally treated at 180 °C for 72 h. After filtrating, washing with water, and calcining in the air at 550 °C for 4 h to remove the organic template, the siliceous MFI zeolite was finally obtained. By NMR analysis, the sample has silanol groups that mostly existed on the zeolite external surface. Therefore, it was denoted as S-1 for distinguishing from S1-OH.

Synthesis of S1-OH-$x$ zeolite ($x$ = 5, 10, 20). The hydroxyl group modified S-1 zeolite was synthesized using DEMS and TEOS as silica sources, and the final products were denoted as S1-OH-$x$, where $x$% was the molar percentage of DEMS to the total amount of silica in the starting gels. As a typical run for the synthesis of S1-OH-5, 80 mL of ethanol was added to 100 mL of a water solution containing 6 mL of aqueous ammonia under stirring. Then, 3.90 g of TEOS and 0.15 g of DEMS were added and stirred at room temperature for another 8 h. After distilling under a vacuum to remove the water and ethanol and drying at 100 °C for 12 h, the solid powder of amorphous silica modified with methyl groups ($SiO_2$-Me) was obtained. The S-1-OH-5 zeolite was synthesized by solvent-free crystallization of $SiO_2$-Me in the presence of TPAOH. After grinding 0.5 g of TPAOH and 0.6 g of $SiO_2$-Me at room temperature for 10 min, the mixture was transferred into an autoclave for crystallization at 180 °C for 72 h to give the

methyl-modified S-1 zeolite. After calcination at 550 °C in air for 4 h, the organic template was removed, and the methyl groups were transformed into hydroxyl groups, which were denoted as S1-OH-5. Furthermore, S1-OH-10 and S1-OH-20 samples were synthesized from the same procedures except using 3.75 g of TEOS with 0.30 g of DEMS, and 3.33 g of TEOS with 0.60 g of DEMS as the silica sources, respectively.

Synthesis of Rh/S1-OH zeolite. Rh/S1-OH was synthesized by an impregnation method. Typically, 1 g of S1-OH zeolite was pre-dehydrated at 200 °C and mixed with 2 mL of $RhCl_3 \cdot 3H_2O$ solution (Rh concentration at 3.9 μmol/mL). Then, the mixture was ultrasonically treated at room temperature for 1 h, followed by grinding the mixture at 60–70 °C to remove the water. The obtained powder was dried at 100 °C for 6 h and calcined at 400 °C for 3 h in air and reduced at 400 °C for 2 h in flowing hydrogen (linear heating to 400 °C, holding for 2 h, 10 vol% $H_2$ in Ar, flow at 60 mL/min) for obtaining the Rh/S1-OH catalyst. The accurate Rh loading in the final Rh/S1-OH zeolite was analyzed by ICP at 0.06 wt%, which is comparable to that in the starting mixture (Rh fraction to the zeolite was 0.08 wt%), suggesting that most of the Rh species have been successfully loaded on the zeolite support.

### Online content

## Data availability
The main data generated in this study are provided in the Supplementary Information. The source data used in this study are available in the Figshare database (https://figshare.com) under the accession code of https://doi.org/10.6084/m9.figshare.22584106. The data that support the findings of this study are presented in the Letter and Supplementary Information are available from the corresponding authors upon reasonable request. Source data are provided with this paper.

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

## Acknowledgements

This work was supported by the National Key Research and Development Program of China (2022YFA1503502), and the National Natural Science Foundation of China (22288101, 22241801, and U21B20101). We thank Prof. Hui Shi at Yangzhou University for kindly discussion on the kinetic study.

## Author contributions

Y.L. performed the catalyst preparation, characterization and catalytic tests. Z.L., W.C., and A.Z. performed the theoretical calculations and wrote the corresponding part. J.Z., C.W., and H.W. participated in the catalyst characterization. Y.Q. and L.S. provided helpful discussion and compiled the process package. Y.H. performed in situ olefin-sorption FTIR characterization. X.Y. performed the NMR characterization. L.W. and F.-S.X. designed the study, analyzed the data and wrote the paper. All authors discussed the results and commented on the manuscript.

## Competing interests

This work has been protected by a Chinese patent with the application number 202110715110.6. The authors Y.L., L.W., and F.-S.X. were involved in this patent. The other authors declare no competing interests.
