## [Peer Review File · Nature Communications]

Rhodium nanoparticles supported on silanol-rich zeolites beyond the homogeneous Wilkinson's catalyst for hydroformylation of olefinsREVIEWER COMMENTS

Reviewer #1 (Remarks to the Author):

This manuscript described the catalytic performance study of Rh loaded silicalite-1 zeolite in hydroformylation reaction. The authors loaded Rh into different supports by wetness impregnation method and characterized these catalysts by XRD, ICP, physisorption, chemisorption, TEM, XPS, SSNMR, FTIR, etc. The characterizations appear to be comprehensive. The author tested their catalysts in hydroformylation of alkenes and found higher TOF than prior literature reports. The significance of this work heavily depends on the accuracy of the TOF. However, there seems to be a serious problem with Rh loading in their catalysts. According to the synthesis method, the theoretical Rh loading should be 0.8 wt%. Since the authors completely removed the solvent, so the actual loading should not be too lower than this. However, the measured loading in SI Table 1 is all around 0.06 wt%, which is highly unlikely. If the authors misplaced the decimal here and the actual loading is around 0.6 wt%, then the reported TOFs will be around ten times lower than the authors claimed values. Therefore, the authors need to clarify this and show detailed calculations in the SI before further evaluations by the reviewers.

Reviewer #2 (Remarks to the Author):

This is a very interesting paper that clearly demonstrates Rh nanoparticles supported on Silanol nests in porous silicate frameworks are very active and selective for hydroformylation. The reactivity of Rh supported on various supports is compared for styrene hydroformylation, including comparisons to homogeneous systems, highlights the unique reactivity of this catalyst. Further, the concentration of silanol nests was various through multiple synthetic approaches, further substantiating the primary conclusion. Further, it was shown that the distinct reactivity is seen for many substrates. Through comparison of reaction orders and via adsorption experiments/modelling, it is argued that strong interactions between the alkenes and silanol nests works to increase the alkene pool around Rh nanoparticles and promote reactivity. The paper is interesting and can be published in close to its current form after a few comments are addressed.

1. It is surprising that the CO reaction orders are very close to 1, as CO strongly interacts with Rh surfaces (> 2 eV binding energies). Is the solubility of CO low in the solvent? Gas phase hydroformylations studies usually show close to -1 reaction order (see recent works from Jingguang Chen et al or Phillip Christopher et al). Some comments on the difference between gas and solvent phase systems could be interesting to connect the results presented here to other recent literature.
2. The DFT calculated adsorption energy of ethylene in silanol nests is small (0.6 eV) as compared to for example CO adsorption energies on Rh (see comment above). So it seems hard to understand how the alkene would outcompete CO for Rh sites (as suggested by the lower rxn order for ethylene with respect to CO) based on what is presented about these systems.
3. Can the authors comment on the >1 rxn order seen for CO? It has been proposed in certain cases that Rh is coordinated by multiple COs to facilitate CO insertion into the Rh-alkyl bond. Is this the cause?

Reviewer #3 (Remarks to the Author):

The manuscript by Liu et al is reporting a new approach for hydroformylation reactions by using tailor made zeolites with dedicated pore cavities called silanol nests. The resulting catalytic activity for both short alkene (ethylene) and aromatic alkene (styrene) hydroformylation is certainly impressive. With such high activity and the ease of recycling by simple mechanical unit operations like filtration, such catalysts become extremely valuable for more benign chemical processes. Given the fact that hydroformylation alone is on a > 10 Mio tons per annum scale, this would constitute a significant benefit.

The overall appearance of the manuscript, its detailed catalyst characterization and detailed investigations ranging from experimental studies to theoretical and simulation is quiet convincing. I have some minor comments:

1. The statement "...using the easily obtained syngas..." is questionable. Most plants which use syngas are world scale size, because the generation of syngas is the most energy-intense and cost-affecting process (see e.g. Fischer-Tropsch synthesis). I suggest skipping the term "easily".
2. In the mechanism study, the authors state that "...apparent reaction orders to styrene,...are well consistent with....where the strong coordination of CO with Rh hinders the olefin interaction..." Does this mean that olefin addition is the rate determining step here? If yes, why do the authors see an apparent order for H₂ of 0.72? That should be zero then.
3. In line with the previous comment: did the authors carry out temperature variation experiments to determine the apparent activation energy? Its value would shed some light on possible transport limitations. As a result, all apparent reaction orders are close to 1, indicative of diffusion effects (always being first order processes).
4. The "pulse-reaction experiment" shown in S31 is looking more like a step-dosing experiment. What is the origin for the high number of peaks? The essence of the experiment is very convincing though.
5. I cannot understand the statement "...such a conversion of 2,4,6-trimethylstyrene...for olefin enrichment." The Rh/SiO₂ catalyst gives identical conversion levels for both styrene and the bulkier 2,4,6-trimethylstyrene. Assuming the pores within SiO₂ being significantly larger than both molecules, this makes sense. But why do the authors relate the lower conversion in the case of Rh/S1-OH, now having pore diameters in the range of the molecule, to the "...crucial role of silanol groups..."? In my opinion, this is just mass transport limitation, or even size exclusion by tailored pore sizes.

The reported concept of enrichment of reactant molecules within certain "pockets" inside porous materials is certainly a highly novel and fundamentally new concept in catalysis. It would be interesting to see whether these catalysts can convert molecules like propene, with an intrinsic regioselectivity problem, in a similar manner. The concept seems applicable to any type of catalysis with active metal centers, hence it is of high interest for the broader scientific community.

Point-by-point responses to the comments

Comments from Reviewer #1

Comments 1-1: This manuscript described the catalytic performance study of Rh loaded silicalite-1 zeolite in hydroformylation reaction. The authors loaded Rh into different supports by wetness impregnation method and characterized these catalysts by XRD, ICP, physisorption, chemisorption, TEM, XPS, SSNMR, FTIR, etc. The characterizations appear to be comprehensive. The author tested their catalysts in hydroformylation of alkenes and found higher TOF than prior literature reports. The significance of this work heavily depends on the accuracy of the TOF.

Responses: Thanks for the comments, highlighting the significance for this work. We have positively and fully responded the comments from the reviewers. These comments really enhance the quality of this manuscript, and thanks for the comments again.

Comments 1-2: However, there seems to be a serious problem with Rh loading in their catalysts. According to the synthesis method, the theoretical Rh loading should be 0.8 wt%. Since the authors completely removed the solvent, so the actual loading should not be too lower than this. However, the measured loading in SI Table 1 is all around 0.06 wt%, which is highly unlikely. If the authors misplaced the decimal here and the actual loading is around 0.6 wt%, then the reported TOFs will be around ten times lower than the authors claimed values. Therefore, the authors need to clarify this and show detailed calculations in the SI before further evaluations by the reviewers.

Responses: Thanks for the comments. Yes, the Rh loading is indeed 0.06 wt% on the Rh/S1-OH sample with high TOFs for the hydroformylation reactions. Following the reviewer's suggestion, we have provided details for calculating the Rh contents and TOFs in the revised SI as follows:

In the synthesis of Rh/S1-OH zeolite, 2 mL of RhCl₃ aqueous solution with Rh concentration at 3.9 μmol/mL was used for the impregnation on 1 g of S1-OH support, which corresponded to Rh weight fraction to the support at 0.08 wt% in the starting mixture, which was obtained as follows:

$$\text{Rh loading content} = \frac{3.9 \mu\text{mol}\cdot\text{mL}^{-1} \times 2 \text{ mL} \times 102.9 \text{ g}\cdot\text{mol}^{-1}}{1 \text{ g}} = 0.08 \text{ wt}\%$$

The accurate Rh loading in the final Rh/S1-OH zeolite was analyzed by ICP at 0.06 wt%, which is comparable to that in the starting mixture, suggesting that most of the Rh species have been successfully loaded on the zeolite support.

The TOF test was performed in a high-pressure stainless-steel autoclave containing a Teflon liner vessel with a total volume of 50 mL. 15 mg of catalyst and 20 mmol of styrene were mixed in 5 mL of toluene solvent in the autoclave. Air in the autoclave was removed by nitrogen, followed by pumping syngas with a molar ratio of CO to H₂ at 1 (molar ratio of CO/H₂/Ar at 45/45/10) three times to remove the nitrogen in the autoclave. The pressure was maintained at a desired pressure at 3.0 MPa, followed by heating to 110 °C to start the reaction. The temperature was measured by a thermocouple inserted into the internal autoclave. After reaction for different periods, the phenylpropyl aldehyde yields were obtained (styrene conversion controlled to be lower than 20%) for calculating the reaction rates. According to the Rh dispersion, the TOF was calculated at 50,306 h⁻¹, which was described as 50,000 h⁻¹ considering the significant digit issue.

We have added this information in the revised manuscript.

Reviewer #2 (Remarks to the Author):

Comments 2-1: This is a very interesting paper that clearly demonstrates Rh nanoparticles supported on silanol nests in porous silicate frameworks are very active and selective for hydroformylation. The reactivity of Rh supported on various supports is compared for styrene hydroformylation, including comparisons to homogeneous systems, highlights the unique reactivity of this catalyst. Further, the concentration of silanol nests was various through multiple synthetic approaches, further substantiating the primary conclusion. Further, it was shown that the distinct reactivity is seen for many substrates. Through comparison of reaction orders and via adsorption experiments/modelling, it is argued that strong interactions between the alkenes and silanol nests works to increase the alkene pool around Rh nanoparticles and promote reactivity. The paper is interesting and can be published in close to its current form after a few comments are addressed.

Response: Thanks for the comments, highlighting the unique reactivity of this catalyst and very interesting results. We have positively and fully responded the comments from the reviewers. These comments really enhance the quality of this manuscript, and thanks for the comments again.

Comments 2-2: *It is surprising that the CO reaction orders are very close to 1, as CO strongly interacts with Rh surfaces (> 2 eV binding energies). Is the solubility of CO low in the solvent? Gas phase studies usually show close to -1 reaction order (see recent works from Jinguang Chen et al or Phillip Christopher et al). Some comments on the difference between gas and solvent phase systems could be interesting to connect the results presented here to other recent literature.*

Responses: Thanks for the comments. Yes, we agree that the kinetic reaction order for CO in the gas-phase reaction is usually negative because of the strong adsorption of CO to the Rh sites, as suggested by the reviewer. However, there are still cases reporting the positive kinetic order to CO in the Rh-catalyzed hydroformylations. For example, Bell and co-workers reported the kinetic reaction order to CO at 1 in ethylene hydroformylation over supported Rh catalysts modified with ligand (*ACS Catal.* 2013, 3, 348); Christopher and co-workers found the kinetic reaction order to bare Rh at -1, but it was changed to 0.6 on the W-modified Rh catalyst (*Nature* 2022, 609, 287). These positive reaction orders to CO were also observed in the other studies (*ACS Catal.* 2013, 3, 2905), which was usually due to the weakened CO adsorption on the Rh sites.

In our case, the reaction was performed in toluene solvent, where the access of Rh sites to CO molecules was obviously hindered, compared with that in the gas-phase system, because of the limited CO solubility in the solvent. Similar phenomenon has been observed previously (*Organometallics* 1995,14, 34; *J. Mol. Catal. A*, 2005, 232, 179).

We have added these data and discussion in the revised manuscript.

Reviewer-Only Figure 1. Kinetic reaction order to (a) CO and (b) H₂ in the hydroformylation of styrene over Rh/S1-OH and Rh/S-1 catalysts.

Note: *The procedures for measuring the kinetic data in hydroformylation.*

As a typical run for measuring the kinetic data of CO, 10 mg of Rh/S1-OH or 20 mg of Rh/S-1 catalyst and 2.5 mmol of styrene were mixed in 5 mL of toluene solvent in the autoclave. 3.0 MPa of the gas containing 0.75 ~ 1.5 MPa of CO (balanced with nitrogen to maintain the total pressure) and 1.5 MPa of H₂ were fed to the autoclave. The data were collected after reaction for different periods, and the reaction rates were obtained according to the linear relationship between conversions (<20%) and reaction time in the tests. The kinetic data for H₂ were performed from similar procedures except for maintaining the pressure of CO and changing the pressure of H₂.

Comments 2-3: *The DFT calculated adsorption energy of ethylene in silanol nests is small (0.6 eV) as compared to for example CO adsorption energies on Rh (see comment above). So it seems hard to understand how the alkene would outcompete CO for Rh sites (as suggested by the lower rxn order for ethylene with respect to CO) based on*

what is presented about these systems.

Responses: Thanks for the comments. Yes, the adsorption energy of ethylene on silanols was lower than that of CO on Rh sites, and the silanol-adsorbed ethylene would not directly compete with the CO adsorption on the Rh. Generally, the kinetic reaction order is a function of the transforming rate to the coverage on the catalyst surface. The olefin enrichment in the catalyst would reasonably reduce the kinetic reaction order from ~1.10 on the Rh/S1 catalyst to close-to-zero on the Rh/S1-OH catalyst. Meanwhile, the CO transformation and adsorption might not be influenced by the different zeolite support, thus leading to similar apparent kinetic reaction orders on both catalysts.

We have added this discussion to the revised manuscript.

Comments 2-4: Can the authors comment on the >1 rxn order seen for CO? It has been proposed in certain cases that Rh is coordinated by multiple COs to facilitate CO insertion into the Rh-alkyl bond. Is this the cause?

Responses: Thanks for the comments. Yes, the kinetic reaction order to CO was ~1.0, which is obviously higher than that in the conventional gas-phase reaction system with negative reaction order to CO. However, there are still cases reporting the positive kinetic order to CO in the Rh-catalyzed hydroformylations (*ACS Catal.* 2013, 3, 348; *Nature* 2022, 609, 287; *ACS Catal.* 2013, 3, 2905). In the reaction in solvents, the kinetic reaction order to CO was positive (*Organometallics* 1995, 14, 34; *J. Mol. Catal. A-Chem.* 2005, 232, 179), which might be due to the solubility issue of CO with limited access to the Rh sites in the solvent system.

In addition, we highly agree that the reaction proceeds with the Rh coordinated by multiple COs, which facilitate CO insertion into the Rh-alkyl bond. This step might not be influenced by the toluene solvent. The reduced CO coverage and unchanged CO transformation would increase the apparent reaction order. With enhancing CO partial pressure to some extent (< 1.5 MPa), the solubility was linearly enhanced to promote the reaction, thus resulting in the first-order phenomenon in the hydroformylation reactions.

We have added such explanation in the revised manuscript.

Reviewer #3

Comments 3-1: The manuscript by Liu et al is reporting a new approach for hydroformylation reactions by using tailor made zeolites with dedicated pore cavities called silanol nests. The resulting catalytic activity for both short alkene (ethylene) and aromatic alkene (styrene) hydroformylation is certainly impressive. With such high activity and the ease of recycling by simple mechanical unit operations like filtration, such catalysts become extremely valuable for more benign chemical processes. Given the fact that hydroformylation alone is on a > 10 Mio tons per annum scale, this would constitute a significant benefit. The overall appearance of the manuscript, its detailed catalyst characterization and detailed investigations ranging from experimental studies to theoretical and simulation is quiet convincing.

Responses: Thanks for the comments, highlighting a new approach for hydroformylation reactions, giving advantages such high activity and the ease of recycling by simple mechanical unit operations like filtration, such catalysts become extremely valuable for more benign chemical processes. We have positively and fully responded the comments from the reviewers. These comments really enhance the quality of this manuscript, and thanks for the comments again.

Comments 3-2: The statement "...using the easily obtained syngas..." is questionable. Most plants which use syngas are world scale size, because the generation of syngas is the most energy-intense.

Responses: Thanks for the comments. Yes, we agree that the production of syngas is a costly process, but it can be still obtained from a wide scope of feedstocks, such as coal, biomass, natural gas, and even plastic wastes. Accordingly, we have modified the sentence as "...using the syngas that can be obtained from coal, biomass, natural gas, and even plastic wastes..." in the revised manuscript.

Comments 3-3: In the mechanism study, the authors state that "...apparent reaction orders to styrene,...are well consistent with...where the strong coordination of CO with Rh hinders the olefin interaction..." Does this mean that olefin addition is the rate determining step here? If yes, why do the authors see an apparent order for H₂ of 0.72? That should be zero then.

Responses: Thanks for the comments. Yes, the CO insertion has usually been regarded as a rate control step in the Rh-catalyzed hydroformylation reactions. However, it is worth noting that the apparent order could not directly represent the rate control step. The positive reaction orders to hydrogen have been observed previously over the phosphine-modified Rh catalyst (*ACS Catal.* 2013, 3, 348), which also has the first reaction order to CO and olefins.

The reduced coverage of molecules on the catalyst surface would have higher apparent kinetic reaction order. In toluene solvent, the hydrogen suffers from poor solubility (*J. Chem. Eng. Data* 1985, 30, 3, 269–273; *J. Mol. Catal. A-Chem.* 1997, 115, 247-257) with relative lower coverage relative to that in the gas-phase reactions. Therefore, the reaction over both zeolite-supported Rh catalysts exhibited positive reaction order to hydrogen in this work.

We have added such information in the revised manuscript. To avoid misunderstanding, we have changed the sentence as follows:

...In the kinetic study in styrene hydroformylation, the Rh/S-1 zeolite catalyst showed apparent reaction orders to styrene, H₂, and CO at ~1.10, ~0.53, and ~0.88, respectively. These results are slightly different from that in the gas-phase hydroformylation, which might be due to the hindered access to the active site in the toluene solvent. For example, the kinetic reaction order for CO in the gas-phase reaction is usually negative because of the strong adsorption of CO to the Rh sites, but the positive reaction orders were obtained in the reaction in solvents...

Comments 3-4: *In line with the previous comment: did the authors carry out temperature variation experiments to determine the apparent activation energy? Its value would shed some light on possible transport limitations. As a result, all apparent reaction orders are close to 1, indicative of diffusion effects (always being first order processes).*

Responses: Thanks for the comments. Yes, we have carried out temperature variation experiments to determine the apparent activation energy of the catalyst. The styrene hydroformylation apparent activation energy of Rh/S1-OH and Rh/S-1 are 51.6 KJ/mol and 49.3 KJ/mol, which are similar to those in the general hydroformylation reactions (*ACS Catal.* 2019, 9, 10899-10912; *ACS Catal.* 2013, 3, 348-357; *Nature* 2022, 609, 287).

Yes, we completely agree that there is a strong diffusion effect in the presence of solvent for the hydroformylation. In this case, the reaction in the liquid phase usually exhibited quite different kinetic reaction orders, which should be reasonably due to the diffusion effects of gaseous molecules, as suggested by the reviewer.

We have added these data and discussion in the revised manuscript.

Reviewer-Only Figure 2. Arrhenius plots of styrene hydroformylation over Rh/S1-OH and Rh/S-1 catalysts. The apparent activation energies for the catalysts are shown in the figure. Reaction condition: 3 MPa of syngas with molar ratio of CO to H₂ at 1 (molar ratio of CO/H₂/Ar at 45/45/10), 2.5 mmol of styrene were mixed in 5 mL of toluene as solvent, 30 mg of catalyst. Styrene conversion was controlled below 20 %.

Comments 3-5: *The “pulse-reaction experiment” shown in S31 is looking more like a step-dosing experiment. What is the origin for the high number of peaks? The essence of the experiment is very convincing though.*

Response: Thanks for the comments. Yes, the experiment in Figure S31 was obtained from a step-dosing experiment. During the tests, the ethylene was pulsed multiple times into the catalyst with a continuous hydrogen flow, and the MS detector gave the signals in the effluent after each pulse experiment, thus resulting in multiple peaks. For example, in the test without CO, the ethylene hydrogenation easily occurred, resulting in the appearance of ethane signal (*m/z* at 30) and reduce signal of hydrogen (*m/z* at 2, negative relative to the background). When CO was introduced with hydrogen, the ethane was undetectable when ethylene was pulsed, accompanied by the extremely weak combustion of hydrogen.

These data help to understand the hindered deep hydrogenation with the existence of CO in the feed, which have been added in the revised manuscript.

Comments 3-6: *I cannot understand the statement “...such a conversion of 2,4,6-trimethylstyrene...for olefin*

enrichment.” The Rh/SiO₂ catalyst gives identical conversion levels for both styrene and the bulkier 2,4,6-trimethylstyrene. Assuming the pores within SiO₂ being significantly larger than both molecules, this makes sense. But why do the authors relate the lower conversion in the case of Rh/SI-OH, now having pore diameters in the range of the molecule, to the “...crucial role of silanol groups...”? In my opinion, this is just mass transport limitation, or even size exclusion by tailored pore sizes.

Responses: Thanks for the comments. Yes, we highly agree that the mass transport limitation is very important for this reaction. In this work, zeolite micropore size displayed a crucial role for the transport limitation during the molecular enrichment. For the zeolite-supported Rh nanoparticle catalyst, the Rh nanoparticles were dominantly loaded on the external zeolite surface because of the impregnation method for the preparation.

In the hydroformylation of 2,4,6-trimethylstyrene, the 2,4,6-trimethylstyrene should directly access the Rh nanoparticles, which is insensitive to the zeolite micropores because the size of zeolite micropores is less than the diameter of 2,4,6-trimethylstyrene. In contrast, for the hydroformylation of styrene, the styrene molecules can be enriched into zeolite micropores with internal silanol groups because of the size of zeolite micropores is larger than the diameter of styrene, which could supply enough styrene molecules for the hydroformylation over the Rh nanoparticles. These results support the crucial role of zeolite with silanol groups for accelerating the hydroformylation of olefins with appropriate molecular sizes. For the case of the Rh/SiO₂ catalyst, it is difficult to enrich the reactants of styrene and 2,4,6-trimethylstyrene, therefore giving similar conversions for both styrene and bulky 2,4,6-trimethylstyrene.

We have added these details in the revised manuscript. In order to clarify the manuscript, we have modified the title of the manuscript as “*Rhodium nanoparticles supported on silanol-rich zeolites beyond the homogeneous Wilkinson’s catalyst for hydroformylation of olefins*”.

REVIEWERS' COMMENTS

Reviewer #1 (Remarks to the Author):

The authors addressed my comments.

Reviewer #2 (Remarks to the Author):

The authors somewhat addressed the comments. While I don't agree with all of the statements and responses, the paper is certainly worth publishing in Nature Comm in its current form.

Reviewer #3 (Remarks to the Author):

The authors have taken great care in responding to all comments and critics raised by the three reviewers. This helped the better understanding of the revised manuscript (including SI), which I consider now publishable in its present form.

Point-by-point responses to the comments

Comments from Reviewer #1: The authors addressed my comments.

Response: Thanks for the comment.

Comments from Reviewer #2: The authors somewhat addressed the comments. While I dont agree with all of the statements and responses, the paper is certainly worth publishing in Nature Comm in its current form.

Response: Thanks for the comments.

Comments from Reviewer #3: The authors have taken great care in responding to all comments and critics raised by the three reviewers. This helped the better understanding of the revised manuscript (including SI), which I consider now publishable in its present form.

Response: Thanks for the comments.